# Lifestyle Factors and Genetic Variants Associated to Health Disparities in the Hispanic Population

**DOI:** 10.3390/nu13072189

**Published:** 2021-06-25

**Authors:** Maria Luz Fernandez

**Affiliations:** Department of Nutritional Sciences, University of Connecticut, Storrs, CT 06269, USA; Maria-luz.fernandez@uconn.edu

**Keywords:** Hispanics, chronic disease, obesity, gene variants, lifestyle

## Abstract

Non-communicable diseases including type 2 diabetes mellitus, coronary heart disease, hepatic steatosis, and cancer are more prevalent in minority groups including Hispanics when compared to Non-Hispanic Whites, leading to the well-recognized terminology of health disparities. Although lifestyle factors including inadequate dietary habits, decreased physical activity, and more prominently, an unhealthy body weight, may be partly responsible for this disproportion in chronic diseases, genetic variations also make a substantial contribution to this problem. In this review, the well-recognized obesity problem in Hispanics that has been associated with chronic disease is examined as well as the influence of diet on promoting an inflammatory environment leading to increased cardiometabolic risk, insulin resistance, fatty liver disease, and cancer. In addition, some of the more studied genetic variations in Hispanics and their association with chronic disease is reviewed.

## 1. Introduction

The main focus on the definition of health disparities varies in different countries [1]. For example, the definition in the United Kingdom is occupational level, while it is education in other European countries and race/ethnicity in the United States (US). The latter definition is the one used in this review. Health disparities can be measured by differences in incidence, prevalence, mortality, burden of disease, and other adverse health conditions [2]. The Hispanic/Latino population (referred to as Hispanic hereafter) has a prominent role in the presence of health disparities in the US due to the preponderance of chronic diseases including type 2 diabetes [3], non-alcoholic fatty liver disease (NAFLD) [4], dyslipidemias associated with cardiometabolic risk [5], cancer [6], as well as a more severe Alzheimer’s disease profile [7] compared to other ethnic or racial groups.

Hispanics across America portray different proportions of African, American Indian, and European ancestry as a result of historical interactions with migrants brought in by the slave trades, European settlement, and the indigenous populations [8,9]. This combined ancestry results in specific gene variants in the Latino population that when combined with dietary and lifestyle habits and a high prevalence of obesity predisposes them to chronic disease. In a recent evaluation of genetic ancestry of self-reported Hispanics (n = 8663) [10], the authors documented that Hispanics in US carry 18% Amerindian Ancestry, 65.1% European ancestry, and 6.2% African ancestry. Interestingly, their distribution in the US varies; for example, those with African ancestry are concentrated in Louisiana and other southern states including Georgia and North Carolina, those of European ancestry are mostly located in the East Coast, and those with Amerindian ancestry are mostly located in the southwest and California. In addition, studies have shown that those Hispanics who report being from Mexico or Central American have a higher contribution of the Amerindian ancestry, while Puerto Ricans and Dominicans have higher contribution of the African ancestry and South Americans as an average have higher European ancestry [11].

A prominent contributor to health disparities in Latinos is obesity and its associated comorbidities [12]. The presence of obesity in Hispanics is reported to be alarmingly high in men and women [13] and has been highly associated with type 2 diabetes (T2D) [14], dyslipidemias [5], and non-alcoholic fatty liver disease (NAFLD) [15]. In addition to obesity prevalence, Hispanics do not necessarily follow the dietary guidelines that recommend higher consumption of fruits and vegetables, decreases in saturated fat and sugars, and reduced sodium [16]. Evaluation of healthy dietary index (range from 1 to 110) going from non-healthy to healthy reported an index from 56 in Puerto Ricans to 71 in Mexicans [17]. Hispanics might also be considered less active than other ethnic groups. For example, the self-reported percent of Hispanics that meet the physical activity guidelines is lower than the non-Hispanic white counterparts (33.4% versus 47.6%) [18].

The purpose of this review is to assess the lifestyle factors including obesity and diet, in addition to genetic variations, that have been identified in the Hispanic population, which individually or in combination lead to health disparities. In this paper, type 2 diabetes, NAFLD, heart disease, and cancer are discussed. Figure 1 depicts the interactions between modifiable and non-modifiable risk factors and the most relevant chronic diseases in Latinos where health disparities have been identified.

## 2. Obesity

Although obesity is a multifactorial disorder that requires environmental conditions to manifest (diet, lack of exercise, sleep deprivation, microbiota composition, endocrine disorders, etc.), genetic factors also play a role in the individual propensity to obesity [19]. A well-established fact is that Hispanic children and adolescents have the highest rates of obesity among the different ethnicities in the United States [20]. It is also common knowledge that obesity plays a major role in the development of type 2 diabetes [21] and heart disease [22]. The Hispanic Community Health Study/Health Study of Latinos (HCHS/Sol) reported a 42.9% prevalence of obesity in women and 36.6% in men with extreme obesity in 7.3% of women and 3.5% in men [13]. Interestingly, the longer time living in the United States was strongly correlated with obesity in Hispanics, indicating that a longer time exposed to social and environmental factors influences behavior and risk of obesity [13]. In this study, acculturation was not associated with obesity [13].

Obesity leads to a number of problems that predispose individuals to heart disease including insulin resistance, dyslipidemias, oxidative stress, and inflammation. Metabolic syndrome (MetS), a condition characterized by central obesity, dyslipidemia (high triglycerides and low HDL), hyperglycemia, and hypertension, and by increasing five-fold the risk for T2D and two-fold the risk for heart disease [23], is more common in obese people [24]. Metabolic syndrome parameters including high triglycerides, hyperglycemia, and insulin resistance appear to be more common in Hispanics compared to other ethnicities [25], further documenting the increased risk for heart disease and type 2 diabetes in this population.

It is important to highlight how obesity is a central problem to the heath disparities in Hispanics. Central obesity is a good predictor of hepatic steatosis [26]. Overweight subjects have two times the risk of NAFLD compared to lean subjects [27], and when obesity is present, one out of three patients will develop non-alcoholic hepatic steatosis (NASH), which is a condition that progresses to hepatic cirrhosis and cancer [28]. It is not surprising, as will be discussed later, that obesity is a strong contributor to the high prevalence of NAFLD among Hispanics. Obesity is also associated with a variety of cancer risks [29]. It has been calculated that if the increases in obesity had not doubled since 1980, at least 25% of cancers would have been avoided [30]. Cancer progression has also been associated with obesity to the point that 14% of cancer deaths in men and 20% of women are attributed to obesity [31]. All these associations between obesity and chronic disease highlight its central role in the health disparities observed in the Hispanic population.

## 3. Type 2 Diabetes (T2D)

The economic costs for diabetes represent a burden to the health system in US. An estimated $327 billion cost including direct medical costs and reduced productivity has been reported [32]. Therefore, it is important to understand why Hispanic adults have 80% and children have 5-fold higher rates than non-Hispanic whites (NHW) [33]. As mentioned earlier, obesity is highly prevalent in Hispanics living in the United States, and as a consequence, they are unduly affected by this disease [33]. In the Hispanic population, a 17% prevalence of diabetes has been reported [34]. Further, poor glycemic control has been associated with low income; thus, impoverished populations have more serious consequences associated with diabetes [35].

In addition to low income [36], other variables including level of education, impaired glucose tolerance, and obesity contribute to the prevalence of diabetes among Hispanics. An inverse correlation between diabetes and education has also been established [37], which could also explain the differences between Hispanics and NHW. The HCHS/SOL study also uncovered a linear correlation of BMI and diabetes with a prevalence of 9.8% with BMI lower than 25 and 14.25% with overweight and obese individuals [14]. In addition, rates for impaired glucose tolerance are higher in Hispanics than in NHW [38]. Lastly, an unhealthy diet and decreased physical activity may also have a high contribution to the prevalence of T2D of Hispanics living in US [39,40]. A higher acculturation of Hispanics, featured by lower fruit and vegetable consumption and higher intake of sugar and added fats, is associated with lower diet quality [41].

In genome-wide association studies (GWAS), the transcription factor-7 like 2 gene (*TCF7L2*) was the first locus associated to type 2 diabetes [42]. *TCF7L2* is a transcription factor involved in the activation of Wnt target genes that specifically repress proglucagon synthesis in enteroendocrine cells. *TCF7L2* polymorphisms can increase susceptibility to type 2 diabetes by decreasing the production of glucagon-like peptide-1 (GLP-1) [43] and decreased insulin production. A significant association was found between reduced insulin secretion and single nucleotide polymorphisms, rs7903146-T and rs112255372, in a group of 1040 Hispanics from the Insulin Resistance Atherosclerosis family study [44].

Hispanics are more insulin resistant than NHW (20.2% versus 15.3%) [45]. The thrifty gene hypothesis postulates that a predisposition to insulin resistance might have protected individuals during periods of starvation by reducing the utilization of glucose in tissues such as muscle and favoring utilization by the brain [46]. Sequence variants have also been identified in *SLC16A11*. In a study conducted in 8214 Mexicans and Latin Americans, the variant *SLC16A11* was found to explain 20% of increased T2D prevalence in Mexico [47]. The risk haplotype has four amino acid substitutions and has been identified in East Asians (10%), American Indians (~50%), and in Hispanics. The risk allele has been associated with an increase of 40% in the odds of developing type 2 diabetes [45]. *SLC16A11* encodes a poorly characterized member of the monocarboxilic acid transporter [48]. Findings from this study suggest that increasing the function of *SLC16A11* could be protective against T2D [48]. However, studies in mice were not able to confirm these findings [49], which suggests that more research is required to understand the role of *SLC16A11* and T2D. The multifactorial nature of T2D in Hispanics is presented in Figure 2.

## 4. Heart Disease

Dyslipidemias characterized by elevated plasma triglycerides, low HDL cholesterol (HDL-C), and high concentrations of LDL cholesterol (LDL-C) are well-established biomarkers for increased risk for cardiovascular disease (CVD) [50]. Hispanics frequently present these dyslipidemias across the age spectrum from children [51] to adults [52]. All these lipid abnormalities in addition to other risk factors lead to cardiometabolic risk [53,54]. Latinos have also been identified with a high prevalence of metabolic syndrome [55], which is a condition that doubles the risk for heart disease.

Lifestyle is a key factor in Hispanics for the development of CHD. Raatz et al. demonstrated that there was a positive association between saturated fat intake and body mass index (BMI) [56]. Hispanics tend to consume diets high in saturated fat and carbohydrate [17], which are two conditions that lead to hypercholesterolemia [57] and hypertriglyceridemia, respectively [58]. The total healthy index for protein foods where meats are included goes from 1 to 5, and the value was 4.92 for Hispanics, and the empty calorie foods where sugars are included goes from 1 to 29, and Hispanics have an average of 15.92 [17]. In addition, saturated fat and simple sugars promote cardiovascular risk by mechanisms that are independent of caloric intake and may be related to increases in glycemic index and re-shaping of the gut microbiome [59]; thus, dietary habits play a major role in increasing the risk for heart disease in Hispanics. In addition, and as mentioned earlier, obesity, which has been shown to be a key factor in developing dyslipidemias, metabolic syndrome, and heart disease, is prevalent in this population. Finally, Hispanics do not follow for the most part the physical activity guidelines provided by the American Society, contributing also to CVD risk [17].

Genetic variants also contribute to factors associated with cardiometabolic risk in Latinos. A rare mutation was identified in the gene encoding adiponectin, ADIPOQ, which explains 17% of the variation of this adipokine in a sample of 1240 Hispanic Americans. This mutation was also associated with very low levels of adiponectin (levels approximately 19% of the non-carriers) [60]. Adiponectin is an adipokine secreted by the adipose tissue that plays a major role in protecting against inflammation and increasing insulin sensitivity [61]. Thus, this genetic variation that results in very low plasma levels of adiponectin can not only lead to increased risk for heart disease but also to insulin resistance and the development of diabetes. The *R230C* variant of the ATP-binding cassette transporter A1 (ABCA1) gene has been associated with decreased HDL-C in the Mexican mestizo population [62], which is an important gene variant due to the correlation between low HDL-C and increased risk for CVD. The *R230C* variant could explain the reported low HDL-C that has been observed in Mexican children [51] and adults [63].

## 5. Non-Alcoholic Fatty Liver Disease (NAFLD)

NAFLD, an obesity-related condition associated with 5–10% fat accumulation in the liver, has become a health concern all over the world [64]. NAFLD is a whole spectrum of liver disease that includes hepatic steatosis (fat accumulation), hepatic steatohepatitis (inflammation), and fibrosis and cirrhosis (scarring) [65]. The presence of NAFLD is confirmed by histology and radiology in the absence of alcohol intake [66]. Several conditions are strongly associated with NAFLD including obesity (65–71%), dyslipidemia (57–68%), hypertension (36–70%), and hyperglycemia (≈37%) [67]. NAFLD is commonly considered another feature of metabolic syndrome, since all parameters are associated with this condition [68]. The most prescribed treatment for individuals with NAFLD is weight loss, exercise and dietary interventions, statins, metformin, and other medications that have proven to be beneficial [69,70].

While the prevalence of NAFLD in the United States is overall about 24%, there is a disproportionate distribution across races with African Americans having the lowest prevalence (17%), European Americans having intermediate prevalence (23%), and Hispanics having the highest (49%) [71,72]. In the Hispanic population, NAFLD has also been observed in children and adolescents, and it is much higher than in their non-Hispanic peers [73]. Furthermore, NAFLD appears to have a worse clinical severity in Hispanics [71]. While this condition has a strong association with obesity, genetics seem to play a key role in the development of this disease. Metabolic abnormalities identified in Hispanics with NAFLD include insulin resistance and higher BMI compared to other populations [4].

In a GWAS analysis of associations with NAFLD, the human patatin-like phospholipase domain-containing 3 (*PNPLA3*) was found to have a strong association in a cohort of Hispanic individuals [74]. The *PNPLA3* gene encodes for a protein of 481 amino acids. The variant rs738409 is a cytosine to guanine substitution, encoding for the isoleucine to methionine substitution at position 148 of the protein. This variant is strongly associated with the entire spectrum of liver disease. Functionally, the *PNPLA3* protein is an enzyme with lipase activity toward triglycerides and retinyl esters and acyltransferase activity on phospholipids. After accounting for variations associated with age, gender, and alcohol intake, evidence demonstrated that hepatic steatosis was found to be heritable in Hispanic Americans, as has been previously reported for European ancestry [75]. This reinforces the concept that the development of NAFLD is partially genetically influenced. The variant near or in *PNPLA3* (rs738409) was significantly associated with hepatic steatosis in Hispanic Americans, and the G allele was higher in this population than in European or African American ancestry, and it was consistent with the higher prevalence of this disease in Hispanics [76]. Furthermore, other genes that were found to be associated with hepatic steatosis in African American and European Americans showed no association in Hispanics, further emphasizing the prominent role of *PNPLA3* in hepatic steatosis in this population.

In addition, the biosynthesis of long-chain polyunsaturated fatty acids LC-PUFA and their metabolites is an essential process that if unbalanced could lead to the accumulation of fat and inflammation in the liver [77]. LC-PUFAs derived from *n*-3 fatty acids lead to the formation of phospholipids, leukotrienes, prostaglandins, endocannabinoids, and other bioactive compounds that have key anti-inflammatory and pro-resolving roles [78]. On the contrary, *n-6* LC-PUFAs and their bioactive metabolites are established to mediate pro-inflammatory responses, although this is a controversial topic. For example, it has been shown in multiple studies that substituting saturated fatty acids (SFA) with *n*-6 PUFA results in a lower risk for heart disease due to their well-reported effect in lowering LDL cholesterol [79]. Furthermore, based on pooled individual-level analysis of prospective studies, circulating and tissue concentrations of linoleic acid (*n*-6 family) were inversely associated with reduced cardiovascular risk, and arachidonic acid was not associated with increased risk [80].

However, regarding population studies, current evidence has shown that American Indians retain the ancient haplotype for the *FADS* gene locus, and this ancient haplotype leads to a lower biosynthesis of bioactive metabolites [81]. Therefore, a supplemented diet with *n*-3 fatty acids would likely benefit this population, as it will increase the substrate available for anti-inflammatory and pro-resolving mediator biosynthesis. In contrast, most African Americans possess the derived haplotype at the *FADS* gene locus, where research has demonstrated that this haplotype efficiently synthesizes these LC-PUFA bioactive compounds. However, because the current Western diet has a very high ratio of *n*-6/*n*-3 fatty acids, there is evidence for a greater synthesis of PUFA metabolites derived from *n*-6 as opposed to *n*-3 due to competitive inhibition at the *FADS* enzymes, ultimately resulting in an increased pro-inflammatory environment [77]. Due to the American Indian and African ancestry in Latinos, they might also benefit by higher intake of *n*-3 fatty acids to alleviate NAFLD and the inflammatory-related diseases. A summary of selected gene variants that have been associated with T2D, CHD, and NAFLD in Hispanics is presented in Table 1.

## 6. Cancer

Cancer will become a more prevalent public health problem by 2030 when it is expected to be the leading cause of death in the US. These data translate into an increase of 45% in the cases of cancer across all ages [82]. This problem will be exacerbated in underrepresented populations, contributing further to health disparities, as a result of multiple factors including health care, socioeconomic status, and late diagnosis, which disproportionately affect minority groups including Hispanics [83], with an estimation of 21% of deaths attributed to cancer in this population [84]. Evidence has also shown that the unequal burden in disease among minority populations can be partially explained by genetic background [85]. When compared to NHW, Hispanics have a lower incidence rate of breast, prostate, and lung cancer; however, they present more of those cancers that are associated with infection [86]. In addition, Hispanics tend to have a cancer diagnosis at an advanced stage compared to NHW due to lower access to medical care, lower socioeconomic status, and proper screening [85].

In contrast to prostate cancer [87], which does not have a high incidence in this population, Hispanic women have been identified to have increased mortality with breast cancer and a higher risk of developing the most aggressive types compared to white women [88]. Notably, in 2009, cancer deaths were the leading cause of death in Hispanics [89]. This situation has been related to problems that exacerbate health disparities including a more advanced stage at diagnosis, lower treatment adherence, and limited access to high-quality care [81].

There are still many unknown factors regarding cancer risk in Hispanics. More information is needed regarding the proportion of genetic ancestry in diverse Hispanic populations and how this relates to cancer prevalence [83]. In addition, Hispanic individuals still contribute to a very low percentage in clinical trials carried out in the US with the purpose of understanding the variation in etiology as well as treatment response [90]. However, understanding the gene variants associated with cancer in Hispanics and the socioeconomic factors that exacerbate cancer in this population will not be sufficient to address health disparities. It is of the utmost importance to educate and tailor programs that promote behavioral changes in Hispanics focused on altering the modifiable factors including maintenance of a healthy weight, appropriate diets, and increased physical activity [91].

## 7. Summary and Conclusions

There are several conclusions that can be derived from this review. There are numerous factors that impact health disparities in the described chronic diseases. Low socioeconomic status, with the associated poverty and low education level, in addition to not having appropriate access to health care and early diagnosis, contribute to the severity of the disease and to the increased mortality. An analysis of the gene variants in Latinos explains in part the increased risk for specific diseases including type 2 diabetes, CHD, and NAFLD. The specific gene polymorphisms present in Latinos are not modifiable; however, their influence on health risks could be somewhat blunted by appropriate diets, as in the case of *n*-3 fatty acids and NAFLD, reductions in simple sugars, or replacing SFA with n-6 PUFA, to improve dyslipidemias and therefore the risk for CVD. It should be emphasized that the reduction of obesity rates should be a priority among Hispanics, since the role of excess weight in the exacerbation of chronic disease is clearly established. Future studies should focus on an in-depth evaluation of gene–diet interactions to better understand health disparities in Hispanics. 

## Figures and Tables

**Figure 1 nutrients-13-02189-f001:**
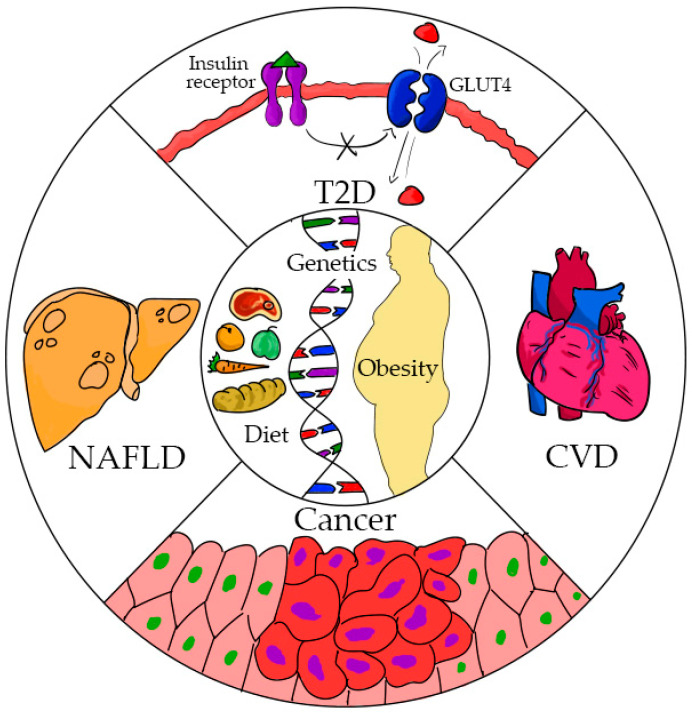
This figure depicts the main components associated with chronic disease in Latinos, gene variants, obesity, and diet (center of the figure) and the most common health disparities in the Hispanic population: non-alcoholic fatty liver disease (NAFLD), type 2 diabetes (T2D), cardiovascular diseases (CVD), and cancer.

**Figure 2 nutrients-13-02189-f002:**
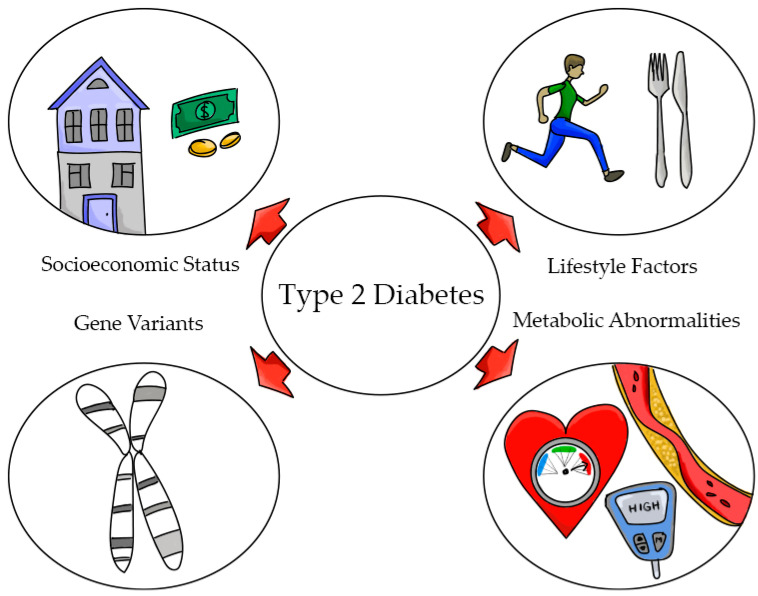
The development of type 2 diabetes is multifactorial in Hispanics. The high prevalence of the disease can be explained by socioeconomic status including low income, lack of education, acculturation; lifestyle factors including unhealthy diets and limited physical activity; and gene variants (*TCF7L2, SLC16A11*) and metabolic irregularities, including obesity, insulin resistance, and dyslipidemias.

**Table 1 nutrients-13-02189-t001:** Selected gene variants associated with type 2 diabetes (T2D), cardiovascular disease (CVD), and non-alcoholic fatty liver disease (NAFLD) in Hispanics.

Disease	Genes	Allele	Protein	Function
**T2D**				
	*TCT7L2*	rs7903146-T, rs112255372	Transcription factor	Involved in the production of wnt genes. Decreased production of glucagon-like peptide-1
	*SLC16A11*	rs77086571	Monocarboxylate transporter	Transport monocarboxylic acids via a proton coupled mechanism
**NAFLD**				
	*PNPLA3*	rs738409	Enzyme with lipase activity	Hydrolyzes triglycerides and retinyl esters
**CVD**				
	*ADIPOQ*	rs200573126	Adiponectin	Low adiponectin levels leads to inflammation and insulin resistance.
	*R230C*	Rs9282541	ABCIA1	Involved in reverse cholesterol transport and in HDL cholesterol levels

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
