# Peer review of "Lifestyle Factors and Genetic Variants Associated to Health Disparities in the Hispanic Population"

_nutrients, 2021, doi:10.3390/nu13072189_

Round 1
Reviewer 1 Report
Please see attached file.

Author Response
In reponse to the reviewer. My responses are in bold. All changes in the revised manuscript are highlighted
Overall: This review addresses an important topic in human nutrition.
R. I thank the reviewer for their helpful comments to improve clarity in this manuscript and I hope I answered all concerns below.
That said, the review needs to be expanded with more detail. As written, it is fairly general in nature. For example, the author states several times that Hispanic populations have a greater risk than non-Hispanic Whites or that a polymorphism increase risk. How much greater risk, what are the quantitative values? More detail is needed for the reader.
R. The percentage of the diseases compared to non-whites are indicated in the corresponding sessions. For example, obesity rates are in lines 66-67. Type-2 diabetes is 5 fold greater in Hispanics compared to NHW (lines 95,97), NAFLD is also much higher in Hispanics (lines 175) and cancer death rates has been added (line 223). All of these have been highlighted to make it easier for the reviewer to find them. In the case of cancer, the numbers are not different from Caucasians but the delay in health care makes the severity of the disease worse for Hispanics.
A section summarizing where future studies are needed and potential barriers would strengthen the paper. Such a section will greatly benefit the readers by provided a resource for thought and planning.
R. What is really lacking is more studies on gene-diet interactions among Hispanics to better understand heath disparities. A paragraph in that regard has been added in the conclusion. The barriers for ameliorating these problems are mostly socioeconomic including poverty and educational levels.
Specifics:
Line 52: More quantitative detail regarding dietary differences is needed. Are these data from NHANES? How much of a difference in fruit and vegetable intake?
R. The Data is from the Hispanic community Health Study/Study of Latinos. More information regarding the diet have been added. Lines 50-52.
Paragraph from line 93 to 103. It seems that paragraph can be incorporated in the related sections of NAFLD, cancer, insulin resistance, etc.
R. I prefer to leave this paragraph where it is. What I tried to do in this paragraph is highlight how obesity, which is a central problem in Hispanics correlates with disease. Later, I discuss all the chronic diseases on their own. The purpose of this sentence was just to focus on obesity. To make this more understandable, I have added one sentence in the beginning.
Line 106: “$” 327 billion
R, Has been corrected
Line 111: The influence of socio-economic conditions is implied here, but not directly stated with respect to Hispanic individuals. This point should be better developed.
R. This is related to the Hispanic population and the first sentence now starts with “In the Hispanic population” to clarify this point.
Line 122: The point of diet and physical activity was already made above.
R. Yes, but in this paragraph, it is directly related to type-2 diabetes since it is well known that physical activity reduces insulin resistance and may improve dyslipidemias in individuals diagnosed with type-2 diabetes as indicated by references 42 and 43.
Line 134. Hispanics are more insulin resistant than NHW. Do you mean by degree e.g. larger HOMA scores, or by percent population? For either, by what degree?
R. The percentage is now included in line 116 and a reference has been added.
Paragraph Line 165 to 175. More quantitative data are needed.
R. Quantitative data have been added, lines 145-147.
Paragraph line 230-249. It is not clear why this paragraph is in this section with NAFLD. I agree that it is am important topic.
R.I believe that this is an important paragraph because NAFLD is highly prevalent in Latinos and it has been shown that those that have the Amerindian ancestry and therefore FADS is not effecient in synthesizing the LC-PUFA, therefore, this would also contribute to the problem of NAFLD
Line 235. The n-6 bad, n-3 good argument is outdated and simplistic. Several recent clinical studies have now shown that n-6 intake also reduces CVD risk. Moreover, many n-6PUFA metabolites like prostacyclin are anti-thrombotic and anti-inflammatory and pro-resolving.
R. There is a lot of controversy in the literature regarding omega-3 and omega-6 fatty acids. The reviewer may be right that individuals with the European ancestry may not benefit so much from omega-3 fatty acids but when one carefully evaluates what happens to minority populations, it is very clear that they do benefit from omega-3 fatty acids. For example, Afro-Americans do benefit from omega-3 fatty acids as they reduce heart disease and in addition, they may protect more Hispanics that have the Amerindian and African American ancestries. In other words, more studies with minority populations would be very helpful in clarifying how certain populations do benefit from omega-3.
Section on Cancer. Again, quantitative detail is needed.
R. The rate of death by cancer among Hispanics has been added (line 223)
Reviewer 2 Report
The review is very well written and the subject is of interest. However I miss the studies carried out in Hispanics on how the diet modulates some of the genetic variants stated. I would encourge including this approach to improve the quality of the work.
Please correct the following mistakes:
Line 82 please correct hight
Line 228 pop-ulation
Reference 90, please remove 1 in the first author's name
Author Response
The review is very well written, and the subject is of interest. However I miss the studies carried out in Hispanics on how the diet modulates some of the genetic variants stated. I would encourge including this approach to improve the quality of the.
R. There are not studies relating diet-gene interactions in Hispanics except for the studies on NAFLD that I already mention in this review (lines 180-207). I point this out as something that needs to be done in future studies in the conclusions (highlighted).
Please correct the following mistakes:
Line 82 please correct hight
R. It has been corrected
Line 228 pop-ulation
R. The program in Nutrients (the journal) does those changes. It cuts the words in unexpected places. I do not know how to fix this
Reference 90, please remove 1 in the first author's name
R. Done
Reviewer 3 Report
The article is interesting and gives an important message about Lifestyle factors and Genetic Variants in the Hispanic Population in America.
A few limitations should be mentioned: The English will require minor revisions. To take some examples from the first two pages: 37 ( for should be to), 40 (lower case a in ancestry), 44 and 46 (remove the), 56 (replace the with their), 72 and 74 ( should be a longer time).
Please revise the second sentence in the Introduction because it implies that there is only one definition in each area discussed.
Only a limited number of polymorphisms were mentioned in this review for the Hispanic population. I think more might be added.
I advise this article should be published with minor revisions.
Author Response
The article is interesting and gives an important message about Lifestyle factors and Genetic Variants in the Hispanic Population in America.
R. I thank the reviewer for providing insightful ideas to improve the content of this manuscript.
A few limitations should be mentioned: The English will require minor revisions. To take some examples from the first two pages: 37 ( for should be to), 40 (lower case a in ancestry), 44 and 46 (remove the), 56 (replace the with their), 72 and 74 ( should be a longer time).
R.Those mistakes have been corrected. The paper was read by a native English speaker whose area of research is genetics. I was told that it is correct to use “the” before African and Amerindian ancestry therefore I have decided to keep those articles.
Please revise the second sentence in the Introduction because it implies that there is only one definition in each area discussed.
R. I am not sure what the reviewer wants me to do with this sentence, maybe that each country highlights more certain aspects of the health disparities therefore I have slighlty modified the sentence to reflect these thoughts. I hope this is what the reviewer meant.
Only a limited number of polymorphisms were mentioned in this review for the Hispanic population. I think more might be added.
R. Those were the polymorphisms where I could find enough information to include in this review. Other polymorphisms are either not well studied or controversial.
Round 2
Reviewer 1 Report
The authors rebuttal to my comment (original version line 235) is not satisfactory. Without citation of differences in dietary intake (including saturated fatty acids and other dietary factors (e.g. simple carbohydrates) that lead to obesity-related comorbidities, it too simple and misleading to make the n-6 bad n-3 good argument. Moreover, n-6 PUFA do lead to the synthesis of anti-inflammatory and anti-thrombotic products like PGI2. In order that the readers are able to examine both sides of the argument, the author should cite the alternate viewpoint regarding the beneficial impact of n-6 PUFA - see Maki et al https://doi.org/10.1093/advances/nmy038, and Marlund et al . 2019 May 21;139(21):2422-2436.
Author Response
I thank the reviewer for bringing up this important point that I overlooked in the first revision. The changes in this version have been highlighted.
The authors rebuttal to my comment (original version line 235) is not satisfactory. Without citation of differences in dietary intake (including saturated fatty acids and other dietary factors (e.g. simple carbohydrates) that lead to obesity-related comorbidities, it too simple and misleading to make the n-6 bad n-3 good argument. Moreover, n-6 PUFA do lead to the synthesis of anti-inflammatory and anti-thrombotic products like PGI2. In order that the readers are able to examine both sides of the argument, the author should cite the alternate viewpoint regarding the beneficial impact of n-6 PUFA - see Maki et al https://doi.org/10.1093/advances/nmy038, and Marlund et al . 2019 May 21;139(21):2422-2436.
Response: I have added one paragraph on lines 201-206 to highlight the beneficial effects of n-6 fatty acids in protecting against heart disease plus I added two references (82 and 83) and in the conclusion on line 253, I acknowledge the role of replacing SFA with n-6PUFA in decreasing the risk for heart disease